# Ethanol Extract of *Campsis grandiflora* Flower and Its Organic Acid Components Have Inhibitory Effects on Autoinducer Type 1 Quorum Sensing

**DOI:** 10.3390/molecules25204727

**Published:** 2020-10-15

**Authors:** Juanmei Zhang, Fenghua Xu, Lingling Yao, Leyu Wang, Miao Wang, Gang Wang

**Affiliations:** 1School of Pharmaceutical, Henan University, Kaifeng 475004, China; Zhangjm@henu.edu.cn (J.Z.); xfh425xfh@163.com (F.X.); yaoll1431103011@163.com (L.Y.); 18300684730@163.com (L.W.); 2Institute of Microbial Engineering, Laboratory of Bioresource and Applied Microbiology, School of Life Sciences, Hennan Univeristy, Kaifeng 475004, China; 3School of Life Science, Hennan Univeristy, Kaifeng 475004, China

**Keywords:** quorum sensing, *Campsis grandiflora*, biofilm, swarming, malic acid, succinic acid

## Abstract

Chinese herbs are a useful resource bank for natural drug development, and have attracted considerable attention to exploit quorum sensing inhibitors (QSIs). This study was designed to screen QSIs from raw Chinese herb materials. Of the 38 common herbs examined, the ethanol extract of *Campsis grandiflora* flower had the strongest QSI activity. The *C. grandiflora* flower ethanol extract (CFEE) was purified by HPD600, and the QSI activities were examined in further detail. CFEE inhibited violacein production of *Chromobacterium violaceum* 026 in a dose-dependent manner, and inhibit the swarming abilities of *Escherichia coli* K-12 and *Pseudomonas aeruginosa* PAO1. Furthermore, CFEE could inhibited biofilm formation and destroyed mature biofilms of *E. coli* K-12 and *P. aeruginosa* PAO1. The composition of CFEE was determined by UPLC-MS/MS to distinguish active QSI compounds, and 21 compounds were identified. In addition to gallic acid and caffeic acid, two organic acids, malic acid and succinic acid, were confirmed for the first time to have autoinducer type 1 QSI activities. Therefore, CFEE is a potential QSI that could be used as a novel antimicrobial agent and should be considered for medicinal development.

## 1. Introduction

Since their discovery, antibiotics, have played a vital role in the treatment of bacterial infections and diseases in humans and animals [1]. However, the emergence and spread of bacterial drug resistance means that new antibiotics are needed to combat drug-resistant bacteria. Only a limited number of new antibiotics have been developed in the past 50 years [2], and infections caused by drug-resistant bacteria are currently one of the leading causes of death worldwide. Without intervention, it is expected that various “superbugs” will emerge one after the other in the future. Therefore, a major challenge for researchers is to develop novel therapies to control diseases caused by these drug-resistant pathogens.

Bacteria can perform coordinated activities, including biofilm formation, swarming motility, and virulence, and this coordination was previously thought to be restricted to multicellular organisms [3,4,5]. Quorum sensing (QS) was an important discovery in the field of microbiology related to the social behavior of bacteria. QS is defined as environmental signal sensing system to monitor population density of bacterial, and regulate a set of bacterial behaviors including bioluminescence, biofilm formation, virulence, swarming motility, competence, antibiotic production, conjugation, and sporulation [6]. Many pathogenic bacteria, including *Escherichia coli* O157: H7, *Pseudomonas aeruginosa*, *Vibrio cholerae*, and *Salmonella enterica* can produce a kind of signal molecule, Acyl hyperserine lactone (AHL), which was named autoinducer type 1 (AI-1), and many phenotypes of them are controlled by this type 1 QS. These phenotypes include the expression of virulence factors, pigment synthesis, and production of degradation enzymes [7,8]. Recent discoveries indicate that intercellular communication enables pathogens to rapidly build up biofilm structures to counter host immunity [9]. In addition, biofilms allow bacteria to resist a variety of extreme environments, including antibacterial substances, high temperature, high salt, and drying [10,11]. Biofilms are therefore novel targets for certain antimicrobial agents [1,6]. Due to their anti-QS activities, quorum-sensing inhibitors (QSIs) can eliminate bacterial pathogenicity or enhance their sensitivity to antibiotics without killing microorganisms, and do not lead to bacteria resistance [12,13]. Thus, it presents a new strategy for the development of new therapeutics against pathogenic bacteria.

AHL is a signal molecule (AI-1) that mediates the QS system of Gram-negative bacteria. Recent researches have focused on the development of AHL analogues as QSIs [14], and many types of QSIs have been reported. Small molecule compounds extracted from plants have attracted significant interest because of their safety and non-toxicity [6,15]. Previous studies on plant polyphenols and flavonoids, such as quercetin, apigenin, and kaempferol, have predominantly concentrated on their antioxidant, anti-inflammatory, and anticancer activities [16,17,18,19]. However, reports of the QSI activities of such compounds have increased rapidly in recent years [20]. Medicinal plants can provide accessible and relatively cheap herbal drugs, and thus, form an integral part of global healthcare worldwide. The broad acceptability of medicinal plants has generated interest in them as a potential source of new, affordable, and efficacious drugs to treat microbial diseases [21]. Chinese herbs are a promising natural medicine resource [22].

Natural compounds from Chinese herbs, including polyphenols, phenolic acids, flavonoids, terpenes, and other herbal ingredients, can be used as QSIs [20]. Previously, we screened natural food materials for active QSI substances, and revealed that aqueous polyphenols from *Rosa rugosa* flower have anti-QS activities [23]. Here, 38 kinds of Chinese herbs were screened for natural compounds with QSI activity. Active substances were selected to verify their effects on AI-1 QS-related bacterial phenotypes, including violacein synthesis, migration, and biofilm formation. This work provides the theoretical foundation for the development of new antibacterial drugs.

## 2. Results and Discussion

### 2.1. QSIs Screening

Thirty-eight flowers from plants used for Chinese herbs were screened using *Chromobacterium violaceum* 026. This reporter strain is a Tn′5 mutant of wild-type *C. violaceum* ATCC 12,472 and cannot synthesize signal molecules itself but can sense exogenous signal molecules. *N*-hexanoyl-l-homoserine lactone (C_6_-HSL) on the surface of the agar plates induces the violacein production of *C. violaceum* 026 in the agar. If QSIs spotted on the filter paper, violacein production will be inhibited, and a transparent, white or opaque ring around the filter paper will appear. The antibacterial activity of the Chinese herb flower extract samples was also detected.

Twelve of the 38 flower extracts, including those from *Campsis grandiflora*, *Sanchi* flower, *Ple butterflybush* flower, and *Flos Sophorae*, could inhibit violacein production of *C. violaceum* 026 (Table 1). Antibacterial activities of these 12 flower extracts were also analyzed. *Sophora japonica Linn* flower, *Chimonanthus praecox*, and *Dendranthema morifolium (Ramat.) Tzvel* displayed antibacterial activities against *C. violaceum* 026. The purple suppressor ring around the filter paper for these three Chinese herb extracts may therefore be due to a bacteriostatic effect of these herbs on *C. violaceum* 026. The other nine flower extracts that inhibited violacein production were considered to be potential QSIs. The crude ethanol extract of *C. grandiflora* demonstrated the strongest anti-QS activity and was selected for further experiments.

*C. grandiflora* flower ethanol extract (CFEE) was purified by macroporous resin HPD600, and the polyphenol content in the extract was determined. Total polyphenol content of the crude ethanol extract from *C. grandiflora* was 247.66 mg GA/g, and that of the CFEE prepared in this study was 480.4 mg GA/g. After purification by macroporous resin, the polyphenols content of CFEE was nearly doubled. This result was likely to be the same as *Rosa rugosa* polyphenols in previous report of Zhang et al. [24].

### 2.2. The QSI Activities of CFEE

QSI activity was checked in the purified CFEE using the agar plate method. When C_6_-HSL is dotted on the filter paper, *C. violaceum* 026 produces violacein to form a purple circle (Figure 1A). While C_6_-HSL was coated on the plate inoculated with 026, the entire plate will appear purple (Figure 1B), and filter papers (Diameter of 6 mm) were placed on the plate to screen QSI. If QSIs are present in the testing sample which were dotted on the filter papers, white or opaque circles will appear around the filter paper on the *C. violaceum* 026 plate. If a sample induces bacteriostasis, the reporter strain cannot grow, and the filter paper will be surrounded by a transparent ring.

As shown in Figure 1B, a ring without purple pigment, approximately 4.27 cm in diameter appeared around the filter paper loaded with CFEE, but there was no transparent ring around the filter paper (Figure 1C). This indicated that the purple suppressor ring was caused by anti-QS phenomenon rather than bacteriostatic action. Therefore, it was concluded that CFEE was a potential QSI and may contain some compounds with QSI activities. Organic extracts from plants were reported to show very promising antibacterial properties correlated with their phenolic compound content [20]. However, due to the developing bacterial resistance, researches on some antibacterial compounds have turned to their anti-QS activities, which can block the bacterial resistance. Medicinal plants contain numerous bio-active molecules, including terpenoids, polyphenols, flavonoids, tannins, anthocyanins, polyamines, cytokinins, and polysaccharides, and these can be useful to counterbalance bacterial resistance by targeting QS signaling pathways [25,26,27,28]. Therefore, the QSI activities of CFEE were further explored, and the small molecule compound composition of CFEE was analyzed.

### 2.3. CFEE Inhibits the Yield of Violacein in C. violaceum 026

The minimum inhibitory concentration (MIC) of CFEE was 1000 μg/mL. All experiments relating to the determination of QSI activities were performed at concentrations below the MIC. To verify the QSI activity of CFEE, violacein production of *C. violaceum* 026 at different CFEE concentrations was determined. The violacein production and biomass were represented as the absorbance of OD595 and OD600, respectively. The yields of violacein did not change significantly (*p* > 0.1) at CFEE concentrations below 200 μg/mL (Figure 2). Increasing CFEE concentrations from 100 to 400 μg/mL, significantly increased the inhibition of violacein production from 5.43% to 67.0% (0.01 < *p* < 0.05). Meanwhile, the biomass of *C. violaceum* 026 was not significantly (*p* > 0.5) inhibited by the concentrations of CFEE (Figure 2). Together, these data indicate that the observed decrease in violacein production is not due to CFEE inhibiting growth of *C. violaceum* 026, but is likely due to CFEE interfering with violacein synthesis, which is regulated by QS.

*C. violaceum* 026 is often used for screening QSIs. When the *C. violaceum* 026 CviR receptor is bound by exogenous C6-HSL, violacein expression regulated by the QS system is activated. Phenolic compounds can inhibit the QS system by competitively binding to the CviR receptor, thereby blocking the production of violacein [29], Thus, inhibition of *C. violaceum* 026 violacein production by CFEE must be the result of active QSI molecules. These molecules require identification and further study.

### 2.4. CFEE Interrupts the Swarming Ability of Tested Strains

To validate the QSI activity of CFEE, the swarming ability of *E. coli* K-12 and *P. aeruginosa* PAO1 under different CFEE concentrations was examined (Figure 3). Swarming circles of the tested bacterial strains were assessed on soft agar plates containing 50, 100, and 200 μg/mL CFEE, respectively; diameters of the swarming circles are shown in Table 2. The diameter of planktonic circle produced by *E. coli* K-12 was 9.02 cm in diameter and covered the entire surface of the plates in the absence of CFEE. However, with increasing CFEE concentrations, the diameter of this planktonic circle decreased significantly (*p* < 0.05). The inhibition rate was up to 90.80% with 200 μg/mL CFEE. In the absence of CFEE, the diameter of planktonic circle formed by *P. aeruginosa* PAO1 was 4.61 cm in diameter, and this diameter also decreased with increasing CFEE concentrations. The inhibition rate of swarming circle diameters in *P. aeruginosa* PAO1 was up to 88.94% with 200 μg/mL CFEE. Therefore, CFEE can interfere with the swarming abilities of both *E. coli* K-12 and *P. aeruginosa* PAO1 without affecting their growth, but the two strains show different degrees of QS interference by CFEE. This result is the same as we previously reported for polyphenols of *Rosa rugosa* [24], and is also consistent with the report by Liu et al. on naringin extract [30]. It is likely that the chemical compositions of extracts from different plants are not identical and may contain different types of polyphenols. Thus, the QSI activities of different extracts may vary, even to the same bacterial strain/species.

### 2.5. CFEE Interrupts Biofilm Formation in Tested Strains

It is generally accepted that the biofilm formation of *C. violaceum* 026, *E. coli* K-12, and *P. aeruginosa* PAO1 are regulated by the QS system. To further verify the QSI activity of CFEE, biofilm yields of these three tested strains were determined (Figure 4).

The total yield of the biofilms formed by *C. violaceum* 026 significantly (0.01 < *p* < 0.05) decreased as the CFEE concentration increased. This indicated that CFEE could inhibit *C. violaceum* 026 biofilm formation and that the rate of inhibition was positively correlated with CFEE concentration. In *E. coli* K-12, the total yield of biofilms did not significantly (*p* > 0.05) change with the increasing CFEE concentrations up to 100 μg/mL. However, CFEE at concentrations greater than 100 μg/mL could inhibit biofilm formation by *E. coli* K-12. CFEE also inhibited biofilm formation by *P. aeruginosa* PAO1 at concentrations greater than 200 μg/mL. In contrast with *E. coli* K-12, the biofilm productions formed by *P. aeruginosa* PAO1 increased significantly (*p* < 0.05) with increasing CFEE concentrations from 25 to 100 μg/mL. These findings indicate that CFEE can interfere with the normal formation of *P. aeruginosa* PAO1 biofilms, which is regulated by the QS system. Although the amount of *P. aeruginosa* PAO1 biofilm increased in the presence of increasing concentrations of CFEE, the structure of the biofilm may have changed. Naringin can loosen the biofilm structure of *P. aeruginosa* PAO1, with an apparent increase in thickness of the biofilm and decrease in density, which is also a typical feature of QSI activity [30]. CFEE may therefore induce changes in the biofilm structure as observed with naringin. Consequently, the scavenging capacity of CFEE on *E. coli* K-12 and *P. aeruginosa* PAO1 biofilms was investigated.

### 2.6. CFEE Scavenging Capacity on Biofilms of Tested Strains

CFEE had a scavenging effect on mature biofilms formed by *E. coli* K-12 and *P. aeruginosa* PAO1 (Figure 5). Untreated by CFEE, mature biofilms had a higher bacterial density and were thicker than those treated with CFEE. When the mature biofilms were treated with 200 μg/mL CFEE, the biofilms become thin and loose. Treatment with 400 μg/mL CFEE almost completely destroyed all mature biofilms formed by both *E. coli* K-12 and *P. aeruginosa* PAO1. This was consistent with early measurements of biofilm productions in this text.

QS influences the initiation and maturation of bacterial biofilms [31]. CFEE inhibited biofilm formation in the tested strains and destroyed mature biofilms. This is congruent with our previous observation of *Rosa rugosa* polyphenols [23]. Therefore, CFEE is a potential QSI for Gram-negative bacteria, and it is worth investigating the active molecule composition and their mechanism of action.

### 2.7. CFEE Chemical Composition

The chemical composition of CFEE was determined using ultra-high performance liquid chromatography (UHPLC)-heated electrospray ionization (HESI)-tandem mass spectrometry (MS/MS). The total ion current of CFEE is shown in Figure 6, and each peak was assigned a number. In the negative mode, 23 peaks were detected. The acquired raw data were analyzed using Compound Discover 3.2 with the metabolite databasemz, and revealed 21 compounds corresponding to 21 chromatographic peaks, respectively. Furthermore, the molecular weights of chromatographic peaks 2 and 10 could be accurately displayed, but the comparison results of the fragment ion information in the database could not infer the compounds.

Details of the 21 compounds identified in CFEE are displayed in Table 3 and characteristic ion fragments of compounds confirmed from each peak are presented in the Appendix A (Figures of MS^1−^ and the fragment ion composition of MS^2−^). Organic acids are one of the principal components of CFEE, with four typical organic acids—quinic acid, glyceric acid, malic acid, and succinic acid corresponding to peaks 4, 6, 7, and 9, respectively. The relative content of these four acids comprise 46.503% of the total of 23 peaks. Additionally, CFEE contains citric acid and a small amount of salicylic acid. Verbascoside (Peak 14), the relative content of which reaches up to 26.438%, is the main polyphenol in CFEE. Further investigation on the QSI activity of this component is recommended for future studies. CFEE also contains many phenolic acid compounds, including caffeic acid, gallic acid, ferulic acid, coumaric acid, and isophthalic acid. Caffeic acid is one of the principal CFEE components, and has previously been reported to be a QSI [32].

While most of the phenolic acid compounds in CFEE have previously been reported to be AI-1 type QSIs [15], QSI activities of organic acids have rarely been reported. Organic acids are one of the principal components of CFEE and are important raw materials used in the food, medicine, and chemical industries. Therefore, organic acids were selected for further analysis and identification of potential new QSIs.

### 2.8. QSI Activities of Malic Acid and Succinic Acid

Malic acid and succinic acid are two very cheap and readily available organic acids. Therefore, the MIC and anti-QS activities of these two compounds were directly tested. The MIC of succinic acid and malic acid in *C. violaceum* 026 were 1200 and 800 μg/mL, respectively. Anti-QS activities of the two compounds were tested at concentrations lower than 800 μg/mL (Figure 7). Non-purple rings appeared around the filter papers loaded with malic acid and succinic acid, indicating that violacein production was inhibited. The anti-violacein circle diameters of CFEE (Figure 7B, 1), malic acid (Figure 7B, 2) and succinic acid (Figure 7B, 3) were 1.89 ± 0.16 cm, 3.06 ± 0.11 cm, 4.67 ± 0.23 cm, respectively. This suggested that the QSI activities of malic acid and succinic acid were stronger than that of CFEE, with succinic acid displaying the strongest QSI activity. These findings indicate that CFEE-mediated inhibition of violacein production is due to the polyphenol, malic acid, and succinic acid contents of CFEE.

UPLC-MS/MS results in this study revealed that organic acids are one of the principal components of CFEE. None of these organic acids have been reported to have QSI activity, except malic acid which has been reported to have inhibitory effect on AI-2 QS system of *E. coli* O157:H7 and *S. typhimurium* [33]. Our results demonstrate for the first time that malic acid and succinic acid have inhibitory effects on the AI-1 QS system of *C. violaceum* 026. Therefore, malic acid and succinic acid may play major roles in the QSI activities of CFEE and could have potential value as antibacterial compounds.

## 3. Materials and Methods

### 3.1. Materials, Test Strains, Culture Media, and Growth Conditions

*C. violaceum* 026 is a kanamycin resistant double mini-Tn5 mutant of *C. violaceum* ATCC 12472, it can’t synthesize its own C6-HSL, but can respond to C_4_-AHL or C_6_-AHL to produce violacein. It was kindly provided by Professor Robert J. C. (Texas State University, San Marcos, TX, USA). *C. violaceum* 026 was routinely cultured aerobically in Luria-Bertani (LB) medium (1% tryptone, 0.5% yeast extract, 0.5% NaCl) supplemented with 20 μg/mL kanamycin in a shaking incubator (120 rpm) at 28 °C. *P. aeruginosa* PAO1 test organism was purchased from Fisheries Research Institute (Shanghai, China). *E. coli* K-12 was supplied by Nanjing Center for Disease Control and Prevention (Nanjing, China). *P. aeruginosa* PAO1 and *E. coli* K-12 were used to determine the biofilm formation and were cultivated in LB medium at 37 °C without shaking.

Dried flowers (Chinese herb source) used in this study were all purchased from Zhang zhongjing pharmacy (Kaifeng, China).

### 3.2. Crude Extract Preparation from the Chinese Herbal Medicines

Dried samples were pulverized with a mortar. Dry powder (5 g) was dissolved in 80% ethanol and treated by an assistant of intermittent ultrasound. The samples were treated by ultrasound intermittently for 60 min, and the supernatant was placed in a 10 mL centrifuge tube and was centrifuged at 4 °C, 4024× *g* for 20 min. The resulting supernatant was distilled to dry using a rotary evaporator, redissolved in 1 mL methanol, and filtered using an aseptic filter. All samples were stored at −4 °C until use in subsequent experiments.

### 3.3. Screening for QSIs

*C. violaceum* 026 was activated in 5 mL fresh LB medium and cultured at 28 °C for 24 h. This suspension was added to fresh LB medium containing 0.85% agar, and kanamycin was added to a final concentration of 20 μg/mL. Next, 50 μL C6-HSLs (40 nmol/L), used as exogenous signal molecules, was coated on the surface of the solidified medium. The agar plates without C_6_-HSLs were used as negative controls. The testing sample (20 µL) was pipetted onto sterile paper discs (6 mm). These disks were placed on the solidified agar plates and 20 µL sterile methanol was used as a negative control. The reporter plates were incubated at 28 °C for 1–2 days.

### 3.4. CFEE Preparation

CFEE was prepared according to a previously report with slight modifications [23]. Polar macroporous resin was selected as the preparation filler because of the strong polarity of organic acid and polyphenols. The crude *C. grandiflora* extract was filtered, diluted with deionized water, and applied to the glass column (17 × 300 mm) containing activated macroporous resin HPD600 (Qinshi Technology Ltd., Zhengzhou, China). The column was washed with deionized water (200 mL) to remove impurities, and the CFEE was eluted using 80% ethanol with a flow rate of 1 min/mL. The solvent was removed by rotary evaporator and the CFEE was re-dissolved in deionized water. The CFEE sample was then lyophilized (at 108 ± 5 Pa chamber pressure and −83 ± 1 °C cold collector temperature after pre-freezing at −20 °C for 24 h), and stored at −20 °C for further experiments. Lyophilized material was re-dissolved in methanol before use in experiments. 

### 3.5. Determination of Total Polyphenol Content of CFEE

The total polyphenol concentration was analyzed using the previously described Folin–Ciocalteu method [34]. CFEE (1 mL, 100 μg/mL) was mixed with Folin–Ciocalteu reagent (1 M, 0.5 mL) for 3 min, then 3 mL of 7.5% Na_2_CO_3_ was added and the mixture was allowed to react for 2 h with intermittent mixing. The absorbance of the mixture was measured at 750 nm using a U-3210 spectrophotometer (Hitachi, Tokyo, Japan). A standard curve, using gallic acid (methanol solution), was prepared using the same method. 

### 3.6. Determination of the MIC of CFEE

MIC is defined as the lowest concentration which showed inhibition of visible growth of the tested bacterial strains [35]. The MIC of CFEE was determined against the tested bacterial strains using the broth macrodilution method as recommended by the Clinical and Laboratory Standards Institute, USA (2006). Bacterial strains were inoculated into 20 mL LB medium supplemented with diluted extracts at final concentrations ranging from 0.1 to 2.0 mg/mL, and were incubated at their optimum culture temperature for 24 h. The absorbance of the media before and after incubation was measured at 600 nm using a spectrophotometer. Sub-MIC concentrations were selected for anti-biofilm and anti-QS activity analyses.

### 3.7. QSI Assay

*C. violaceum* 026 was used as a reporter strain to measure the QSI activity of CFEE as previously reported [23]. *C. violaceum* 026 was inoculated into 4 mL LB medium and incubated for 18 h at 28 °C. The culture was diluted 1:50 into 4 mL fresh LB medium containing 40 nM C6-HSL and CFEE (25, 50, 100, 200, and 400 μg/mL) and was incubated for 48 h at 28 °C. A sample without CFEE was used as a control. Cultures were vortexed to resuspend any adherent cells, and 300 μL was placed in a 1.5 mL microtube. Cells were lysed by adding 150 μL of 10% sodium dodecyl sulfate (SDS), followed by vortexing for 10 s. Violacein was extracted from the cell lysate by adding 600 μL water-saturated butanol, vortexing for 5 s, and centrifuging at 1611× *g* for 5 min. The butanol phase (upper layer) containing violacein was transferred into 96-well microtiter plates, with 150 μL per well. The absorbance of the extract was measured at 595 nm using a microplate reader (Bio Tek, Winooski, VT, USA). The effects of CFEE were evaluated based on relative violacein production levels, with the control value set to 100%. All experiments were performed in triplicate, and data are presented as means ± standard deviation.

### 3.8. Swarming Motility Inhibition Assay

Swarming motility assays were performed as previously reported [36]. Briefly, 5 mL molten soft-top LB medium (containing 0.3% agar) was mixed with CFEE at final concentrations of 50, 100, and 200 μg/mL, respectively. These mixtures were immediately overlaid on solidified LB agar plates. Once the overlaid agar had solidified, prepared pathogen strains were applied to the center of the plate using an inoculation needle and were then cultivated at 37 °C for 30 h. Swarming migrations were observed and migration distances of *C. violaceum* 026, *E. coli* K-12, and *P. aeruginosa* PAO1 were measured. All experiments were performed in triplicate, and data are presented as means ± standard deviation.

### 3.9. Biofilm Formation Inhibition Assay

Biofilm formation of the studied strains was determined as described previously [37], with some modifications. A total of 600 μL LB medium containing different concentrations of CFEE (0, 25, 50, 100, 200, and 400 μg/mL) was added into a small glass tube, inoculated with 6 μL bacterial strain, and incubated at 37 °C for 72 h. After incubation, the medium in each tube was gently removed with a pipette, and each tube was washed three times with 1 mL distilled water to remove unattached bacteria. Biofilm cells were then stained with crystal violet for 30 min. Excess dye was removed with deionized water and the crystal violet adsorbed by biofilm cells was eluted with a mixture of ethanol and acetone (2:1), and the OD590 of the eluant was measured. Differences in biofilm formation were analyzed by one-way analysis of variance (ANOVA) followed by Tukey′s pairwise post-hoc comparisons.

### 3.10. Determination of the Scavenging Capacity of CFEE on the Mature Biofilms

Overnight cultures of tested bacterial strains incubated in LB medium were diluted to a final density of 1.0 × 10^6^ CFU/mL with fresh media, and 1 mL was dispensed into each well of a 24-well microtiter plate together with a piece of coverslip with 0.2 mm thick and 13 mm diameter (Nunc, Roskilde, Denmark). Plates were statically incubated at 37 °C for 72 h. Coverslips with a mature biofilm of tested strains were then placed in LB medium supplemented with 200 or 400 µg/mL CFEE and statically incubated at 37 °C for 24 h. A group of bacteria without CFEE was used as a negative control. Coverslips were then gently washed with sterile distilled water, stained with safranine, and observed and photographed using a microscopic imaging system.

### 3.11. CFEE Components Assay by UHPLC-MS/MS

Individual phenolic composition analyses were carried out by UHPLC-MS/MS analysis using a Q Exactive Plus mass spectrometer (Thermo Fisher, Waltham, MN, USA) coupled with the vanquish-flex with a Hypersil GOLD column (2.1 × 100 mm, with 1.9 µm coating layer) (Thermo Fisher, Waltham, MN, USA). One microliter of sample was loaded, and the temperature of the column box was set to 30 °C. According to our previous report [38], the mobile phase was A: HPLC grade H_2_O (0.1% formic acid); B: HPLC grade acetonitrile. The chromatographic separation program was: 10% acetonitrile elution for 2 min, then was increased to 50% within 5 min and maintained for 3 min, then was increased to 80% within 10 min, and finally increased to 100% within 5 min. The flow rate was 0.3 mL/min.

The mass spectrometer was operated in full scan of positive and negative mode in a range of *m/z*: 70–1050. The resolution of MS data was set at 70,000; the spray voltage was set at 3.2 kV for negative mode; the capillary temperature was set at 320 °C; the auxiliary gas heater temperature was 350 °C; the sheath gas flow rate was 35 units; the auxiliary gas flow was set at 15 units; the S-lens RF level was 50; the AGC target was 1e6; the maximum injection time used was 100 ms; and the resolution of MS/MS acquisition was set at 17,500. The top eight ions in each full scan were isolated with a 1.0 Da window, fragmented with a stepwise collision energy of 20, 40 and 60 units, and a maximum injection time of 50 ms with an AGC target of 2e5. The acquired raw data were analyzed using Compound Discover 3.2 (Thermo Fisher, Waltham, MN, USA) with the metabolite database (mzCloud, mzVault, Masslists, and Chemspider).

### 3.12. Statistical Evaluations

Vegetative growth curves of the wild-type and mutant strains were generated by plotting the average outcomes (OD600) of three experiments per strain. Differences in biofilm formation were analyzed by one-way analysis of variance (ANOVA) followed by Tukey′s pairwise post hoc comparisons.

## 4. Conclusions

At present, many kinds of natural compounds have been studied to use as new sources of QSIs. Chinese herbs have attracted much attention because of their safety and affordability. A screen of 18 different Chinese herbs for QSI active substances found that, the alcohol extract of *C. grandiflora* flower had strong QSI activity. CFEE could inhibit the yields of violacein produced by *C. violaceum* 026 in a dose-dependent manner. It also inhibited biofilm production by *C. violaceum* 026 and *E. coli* K-12, and the rate of inhibition positively correlated with CFEE concentration. CFEE could also inhibit biofilm formation of *P. aeruginosa* PAO1 at concentrations greater than 200 μg/mL. But it is interesting that CFEE significantly increased the biofilm production of *P. aeruginosa* PAO1 (*p* < 0.05), in a concentration-dependent manner (25 to 100 μg/mL). Although the amount of *P. aeruginosa* PAO1 biofilm increased in the presence of CFEE, the structure of the biofilm changed and became loose. These results show that CFEE can interfere with the normal biofilms formation of the test strains, and can also destroy the mature biofilms of *E. coli* K-12 and *P. aeruginosa* PAO1, indicating that CFEE was a potential QSI that could be considered for medicinal development as a novel antimicrobial agent.

To investigate and identify the active QSI active compounds in CFEE, UPLC-MS/MS was used to determine the composition of CFEE. CFEE contains numerous phenolic acids, such as caffeic acid, gallic acid, ferulic acid, coumaric acid and isophthalic acid, and most of these compounds have previously been reported to be AI-1 type QSIs. Meanwhile, CFEE also contains many organic acids, including malic acid, quinic acid, and succinic acid, whose relative content comprise 46.503% of the total 23 compounds. Additionally, citric acid and a small amount of salicylic acid are present in CFEE. QSI activities have not been reported for any of these organic acids with the exception of malic acid, which has an inhibitory effect on AI-2 QS of *E. coli* O157:H7 and *S. typhimurium*. Further experiments demonstrated that malic acid and succinic acid were excellent AI-1 QSIs. These two organic acids may play a major role in the QSI activities of CFEE. Future research on malic acid and succinic acid will explore their molecular mechanisms and the targets of their anti-QS effects. In summary, CFEE and its malic acid and succinic acid components are potential QSIs that could be used to develop new therapeutics targeting Gram-negative bacteria.

## Figures and Tables

**Figure 1 molecules-25-04727-f001:**
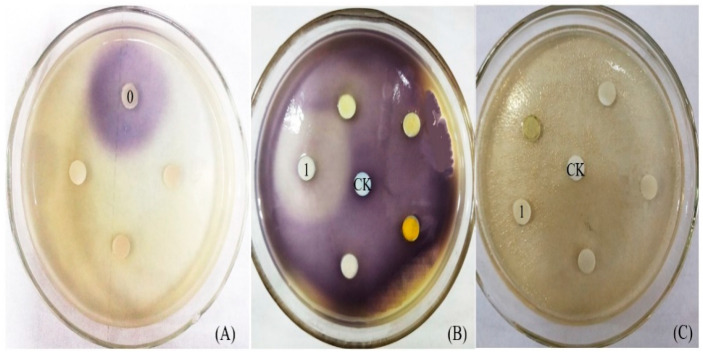
The QSI activity of *C. grandiflora* flower extract and CFEE identified by report plates. (**A**,**C**) LB agar plate incubated by *C. violaceum* 026; (**B**) LB agar plate supplemented by C_6_-HSL and incubated by *C. violaceum* 026; “CK”, negative control loading with methanol; “0”, C_6_-HSL; “1”, CFEE.

**Figure 2 molecules-25-04727-f002:**
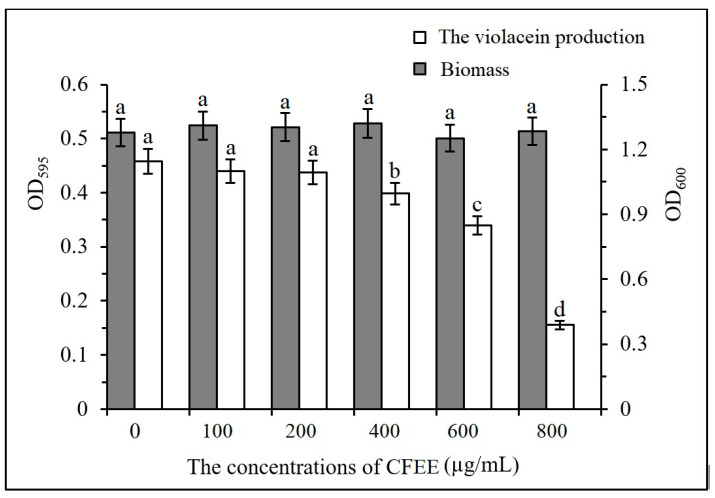
Biomass and violacein production of *C. violaceum* 026 when it was incubated in the LB medium with different concentrations of CFEE. Each bar represents the standard deviations of five measurements. Different letters indicate significant differences in inhibition rates.

**Figure 3 molecules-25-04727-f003:**
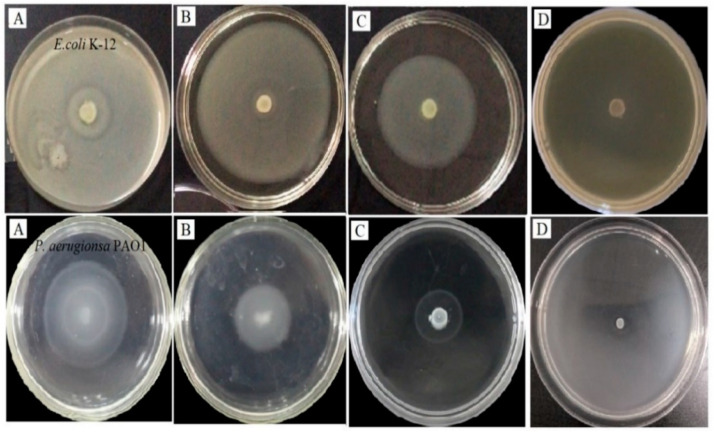
CFEE affected on migration distance of *E. coli* K-12 and *P. aerugionsa* PAO1: (**A**), CK; (**B**), 50 μg/mL of CFEE; (**C**), 100 μg/mL of CFEE; (**D**), 200 μg/mL of CFEE.

**Figure 4 molecules-25-04727-f004:**
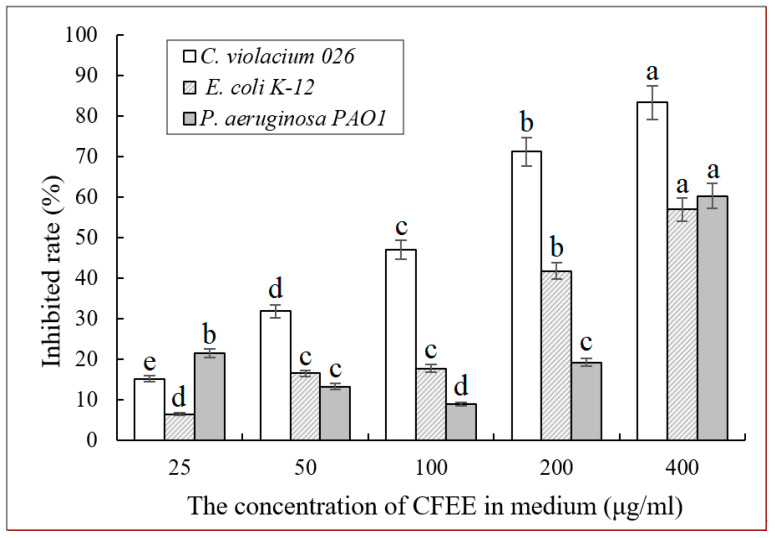
Inhibited rate of bacteria biofilm formation of tested strains under different concentrations of CFEE. Each bar represents the average of five measurements and their standard deviations. Different letters indicate significant differences in inhibition rates.

**Figure 5 molecules-25-04727-f005:**
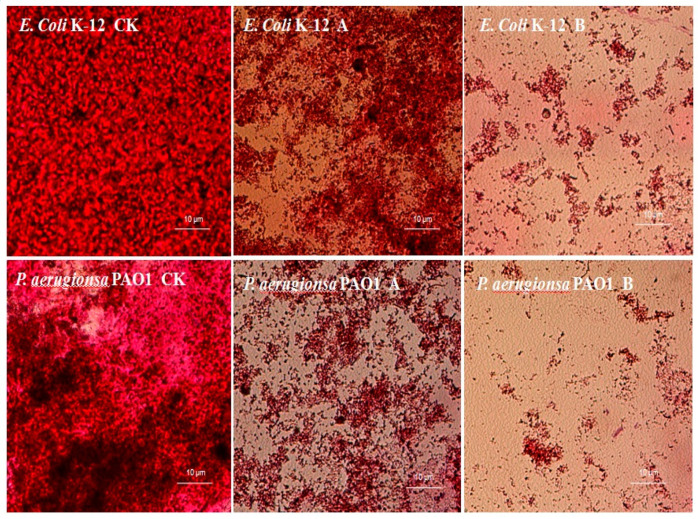
Scavenging effects of CFEE on mature biofilms formed by *E. coli* K-12 and *P. aeruginosa* PAO1. (**A**) 200 μg/mL; (**B**) 400 μg/mL.

**Figure 6 molecules-25-04727-f006:**
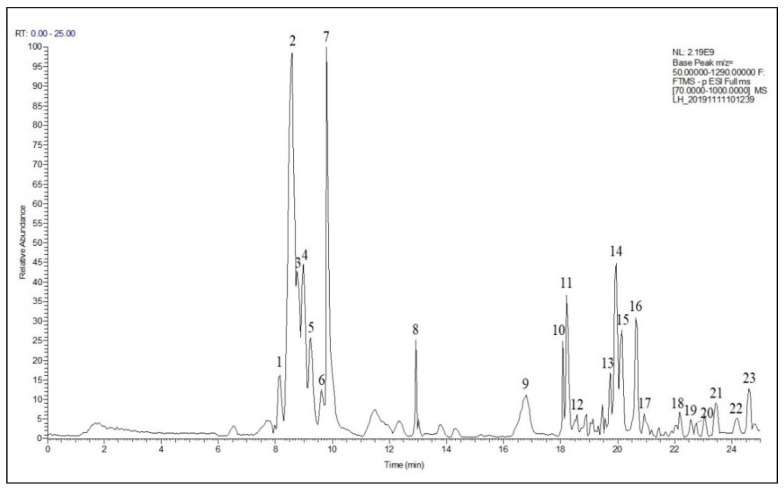
The total ion current of CFEE identified by UHPLC-HESI-MS/MS.

**Figure 7 molecules-25-04727-f007:**
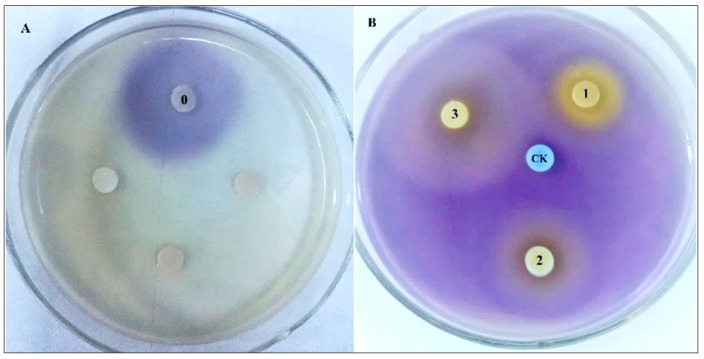
QSI activity of crude *C. grandiflora* flower extract and CFEE identified by report plates. (**A**) LB agar plate incubated by *C. violaceum* 026; (**B**) LB agar plate supplemented by C_6_-HSL and incubated by *C. violaceum* 026; “CK”, methanol as negative control; “0”, C_6_-HSL; “1”, CFEE; “2”, 200 μg/mL succinic acid; “3”, 200 μg/mL malic acid.

**Table 1 molecules-25-04727-t001:** Screening results of 38 kinds of plant flowers used for Chinese herbs.

	Inhibition Zone of Purple	Transparent Zone
*Campsis grandiflora*	+++	-
*Panax notoginseng* flower	++	-
*Buddleja officinalis* flower	++	-
*,Pueraria lobate* flower	++	-
*Lonicera japonica Thunb.*	++	-
*Sophora japonica Linn* flower	+	++
*Crocus sativus L.*	+	-
*Lavandula angustifolia Mill.*	+	-
*Jasminum sambac (L.) Ait.*	+	-
*Chimonanthus praecox*	+	++
*Dendranthema morifolium (Ramat.) Tzvel.*	+	+
*Prunus persica*	++	-

Footer: “+”, inhibition zone diameter under 9 mm; “++”, inhibition zone diameter between 9 and 14 mm; “+++”, inhibition zone diameter higher than 14 mm, “-“, no inhibition zone.

**Table 2 molecules-25-04727-t002:** Inhibition rates of planktonic circle by different concentrations of CFEE.

CFEE Concentration	50 μg/mL		100 μg/mL		200 μg/mL	
	Diameter (mm)	Inhibited Rate (%)	Diameter (mm)	Inhibited Rate (%)	Diameter (mm)	Inhibited Rate (%)
*E. coli* K-12	75.4 ± 0.8	16.41	50.7 ±1.3	43.79	8.3 ± 0.3	90.80
*P. aerugionsa* PAO1	28.7 ± 1.1	37.74	26.5 ± 0.2	42.52	5.1 ± 0.1	88.94

**Table 3 molecules-25-04727-t003:** The compounds in CFEE as identified by UHPLC-HESI-MS/MS.

Peak No.	RT (min)	Molecular Weight	Molecular Formula	Δmass (ppm)	Tentative Identification	mzVault Match	Content (%Area)
1	8.154	132.05	C_4_H_8_N_2_O_3_	−10	Asparagine	95.9	0.620
2	8.585	208.06	C_4_H_4_N_10_O	-	Unknown	-	0.129
3	8.628	194.04	C_6_H_10_O_7_	−3.41	Glucuronic acid	64.0	7.467
4	8.973	192.06	C_7_H_12_O_6_	−5.03	Quinic acid	91.8	15.090
5	9.017	192.03	C_6_H_8_O_7_	−2.91	Citric acid	94.0	2.739
6	9.237	106.03	C_3_H_6_O_4_	−14.31	Glyceric acid	87.3	7.502
7	9.785	134.02	C_4_HO_5_	−10.1	Malic acid *	97.0	16.948
8	12.925	261.08	C_10_H_15_NO_7_	-	Hymexazol O-glucoside	-	0.021
9	16.739	118.03	C_4_H_6_O_4_	−4.27	Succinic acid *	96.2	6.964
10	18.071	-	-	-	Unknown	-	0.198
11	18.204	316.08	C_13_H_16_O_9_	2.96	NP-020139	67.0	0.246
12	18.617	170.02	C_7_H_6_O_5_	−6.16	Gallic acid	-	0.167
13	19.742	356.11	C_16_H_20_O_9_	UN	Gentiopicrin	73.9	0.124
14	19.939	624.20	C_29_H_36_O_15_	−1.88	Verbascoside	92.1	26.438
15	20.134	166.03	C_8_H_6_O_4_	−6.89	Isophthailc acid	81.5	0.190
16	20.622	180.04	C_9_H_8_O_4_	−5.39	Caffeic acid *	94.0	6.377
17	20.924	138.03	C_7_H_6_O_3_	−9.31	Salicylic acid	86.3	1.414
18	22.163	164.05	C_9_H_8_O_3_	−0.14	Coumaric acid	79.4	1.260
19	22.543	194.06	C_10_H_10_O_4_	−3.42	Ferulic acid	94.7	1.276
20	22.706	164.05	C_9_H_8_O_3_	−6.5	Coumaric acid	91.6	0.309
21	23.477	146.04	C_9_H_6_O_2_	-	Coumarin	57.9	0.778
22	24.275	145.05	C_9_H_7_NO	−8.84	4-Indolecarbaldehyde	77.4	0.207
23	24.591	328.22	C_18_H_32_O_5_	−2.74	Corchorifatty acid	89.4	3.537

“*” means identified by using a standard sample.

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
