# Peer review of "Ethanol Extract of Campsis grandiflora Flower and Its Organic Acid Components Have Inhibitory Effects on Autoinducer Type 1 Quorum Sensing"

_molecules, 2020, doi:10.3390/molecules25204727_

Round 1
Reviewer 1 Report
In general, the work is well developed and exposed, there are some modifications that are considered important to clarify the presentation of the results.
- For the results presented in Figure 1, Figure 3 and Figure 7. It is recommended to normalize the results based on "mm of inhibition" or "% inhibition" to improve the discussion of results at the different concentrations studied, in this way significant differences can be established between the different concentrations and substances used.
- In the title of Figure 2 and Figure 4. Add the statistical analysis,
Are the graphs with bar of error or standard deviation?. Specify the type of error, the number of samples analyzed (n) and the p values obtained.
- There is no statistical analysis described that allows improving the discussion of the differences in the production of "C. violacium and biomass" for Figure 2 and also for the influence of P. aeruginosa (PAO1), E. coli (K-12) and C. violacium (026) in the formation of biofilms in Figure 4. Perform and better describe this statistical analysis.
Reviewer 2 Report
Dear Editor and Authors,
Please, find my comments, recommendations, raised questions and technical advices on the manuscript below:
General comment: The present manuscript is dedicated to investigation of quorum sensing inhibitory activities of 80% ethanolic extract from the flowers of Campsis grandiflora. The scientific and practical significance of the manuscript cannot be disputed. However, all sections of the paper need to be improved (see Details) because of some conceptual, technical and result interpretation issues. For example, CFP contains representative amounts of organic acids but it is named a phenolic sample. Furthermore, the whole manuscript has to be checked and corrected by a professional English editor. Therefore, I would like to recommend a major revision before reconsideration for publication in Molecules.
Details:
Abstract:
Lines 11-13: At present, Chinese herbs, an excellent resource bank for natural drug development, has attracted much attention to exploit quorum sensing inhibitor (QSI) because of its security and affordable. – Please, check English.
Line 13: In this text – It is better to use an expression similar to: The aim of the current study
Lines 13-15: Our results show that the alcohol extract of Campsis grandiflora flower during 18 kinds of common herbs has strong QSI activity. - Please, check English.
Lines 15-16: We prepared the C. grandiflora polyphenols (CFP) and determined its QSI activities. – It is better to say that you have extracted the C. grandiflora polyphenols; its QSI activity or their QSI activities
Line 16: inhibit the yields – Please, check English
Line 21: components should be replaced by compounds
Lines 21-22: The principal component is organic acids, such as malic acid and succinic acid, which haven’t been reported to have autoinducer type 1 QSI activities. - Please, check English
Lines 22-23: Therefore, CFP is a potential QSI which could be considered for medicinal development as a novel antimicrobial agent. – Please, check English. In general, CFP should be in a plural form because you do not explain/report the activity of a particular polyphenol compound from the plant.
Introduction:
Line 27: Maybe it is better to use the plural form antibiotics because you explain in general.
Line 28: bacteria-related diseases
Line 30: You should use infections
Line 42: Salmonella enterica
Lines 41-45: Previous reports have shown that many pathogenic bacteria such as Escherichia coli O157: H7, Pseudomonas aeruginosa, Vibrio cholerae and Salmonella enteric could produce a kind of signal molecules, and lots kind of phenotype are controlled by the autoinducer type 1 (AI-1) quorum sensing, such as the expression of virulence 44 factors, pigment
45 synthesis, and expression of degrading enzymes [7-8]. – Please, split the sentence into two new ones and improve English (e.g. lots kind of)
Line 43: could should be replaced by can
Line 56: in the recent years
Lines 56-58: Many types of QSI have been reported, among which, a kind of small molecule compound extracted from natural plants, is of great interest at present because of its safety and non-toxicity[6,15]. – It means that you speak about one compound. If this is not true, please use a plural form in the whole sentence.
Lines 59-61: Previous studies on plant polyphenols and flavonoids mainly focused on their antioxidant, anti-inflammatory and anticancer activities, such as quercetin, apigenin, and kaempferol [16-19]. – Please, check English.
Line 68: that could
Line 72: It is better to replace component with compound
Results and discussion:
Line 79: that could/can
Line 86: activities or activity
Line 96: were
Line 98: as a
Line 105: Please, replace who by which
Line 105-108: The report strain C. violaceum 026 is a Tn’5 mutant of wild-type C. violaceum ATCC12472, who could not synthesize signal molecules by itself, but can make use of exogenous signal molecules, so it can produce violacein while N-hexanoyl-L-homoserine lactone (C6-HSL) was dotted on the filter paper (Figure 1 A). – Please, split it into two sentences.
Line 114: CFP is a potential QS inhibitor – CFP should be used in a plural form, as previously mentioned.
Line 115: components should be replaced by compounds, as previously mentioned.
Lines 121-122: Then, we further determine the QSI activities of CFP, and analyze its composition of small molecule compounds. – Please, reconsider the tense (maybe past simple?)
Line 133: ,while
Line 136: biomass accumulation?
Line 138: does not due to – English
Lines 142-143: Small molecular phenolic compounds – English
Line 145-146: However, what exactly are these QSI active molecules that need further analysis and study. – Please, read this sentence and reformulate it, if necessary.
Line 160: and it is…
Lines 161-163: It is possible that different types of polyphenols and chemical composition of the extracts from different plants, so the interference mechanism for the QS system of the same bacteria is also slightly different. – English
Line 164: CFP – plural form; they have different degrees…
Lines 163-165: Therefore, CFP can interfere with the swarming ability without affecting the growth of E. coli K-12 and P. aeruginosa PAO1, and has different degrees of interference to the QS system of different strains. – What is your explanation/speculation on that observation?
Line 174: formatted?
Line 178: inhibition of yields? – English
Line 179: until its concentration over 200 μg/ml. – English (line 180 – the same)
Line 195: in a good agreement; biofilm yields
Line 204: 2.8. Chemical composition of CFP – You did not mention the result from total phenolic content determination. Why did you name your sample CF polyphenols, when you have representative amounts of organic acids in your sample? Please, reconsider this formulation.
Line 211: There are 21 kinds of chemical composition in CFP – English
Line 212: details
Line 213-214: All characteristic ion fragments of the compound confirmed from each peak were showed in the supplementary document.
Line 214-215: organic acid is the principal component of CFP. – one organic acid?
Line 219: content should be replaced by compound
Line 222: citation
Line 223: Please, reconsider the new paragraph formation (maybe when you begin with phenolic compounds).
Line 224: inhibitors
Line 224: Coumarin has also been reported to have QSI activity [citation]
Line 228: Please, change the title of the table because asparagine, glucuronic acid, etc. are not phenolic compounds.
Question: How did you determine the L and D configurations of some compounds?
p-, trans- in italics
Lines 238-239: These results indicate that CFP inhibited the production of violacein was not only
due to its polyphenols content, but also the content of malic acid and succinic acid. – English
Line 239: polyphenol content
Line 248: Our experimental results have shown for the first time
Line 251: will be potential antibacterial compounds
Lines 245-246: organic acid are the principal component of CFP? You cannot use a singular form when you speak in general for the group/class of organic compounds.
Materials and methods:
Lines 254-257: C. violaceum 026 (a double mini-Tn5 mutant of the wild-type strain C. violaceum (ATCC 31532) with a kanamycin resistance that was unable to synthesize its own C6-HSL, but it retains the ability to respond C4-AHL or C6-AHL and produces violacein was kindly provided by Professor Robert J. C. (Texas State University, USA) which was used as biomonitor strain. – This is too much information for one sentence.
Lines 264-265: Dried flowers (used for Chinese herbal medicine) selected in this experiment were all purchased from Zhang zhongjing pharmacy (Kaifeng, China). – Did you consult with a botanist for confirmation of the plant material?
Lines 268-271: The ultrasonic breakers were crushed 60 min, the supernatant was taken in the 10 ml centrifuge tube, 4℃ and 6000 rpm centrifuged for 20 min. The supernatant was distilled to dry, and 1 ml methanol was dissolved. Finally, the filtrate was filtrated with the aseptic filter. – English
Line 271: next experiments
Line 273: at 30°C; additional space?
Lines 273-277: Then, the suspension was added to fresh LB medium containing 0.85% agar, and kanamycin was added with a final concentration of 20 μmol/l (M), and then, 50 μl C6-HSLs (50 M), used as exogenous signal molecules, was coated on the surface of the solidified medium, and the agar plates without C6-HSLs as negative control. – This is too much information for 1 sentence.
Line 278: used 20 μl deionized methanol as a negative control. – Maybe it can be placed as a new sentence or you need to write it correctly in English in the same sentence.
Line 279: at 30°C
Line 282: extract were?
Line 283: Chromatogram column?
Line 285: impurities
Lines 286-287: Eluted fraction was removed solvent by a rotary evaporator and redissolved with an appropriate amount of deionized water, then, lyophilized. – English
Line 289: power?; re-dissolved in
Line 293: Please, cite also the article with the original method. – 2 h is too much time. There protocols for 5 min at 50 degrees.
Line 303: media
Line 304: a spectrophotometer
Line 312: a control
Line 318: Then, analyzed extract – English
Line 325: Then, poured immediately
Line 326: was solidified
Line 328: migrations were or migration was
Line 330: All experiments; data were
Lines 348-350: Then, took out the coverslip with mature biofilm of tested strains, and put it in the LB medium supplemented with 200 μg/ml and 400 μg/ml CFP, respectively, using a test group without CFP as control. – This sentence sounds like an instruction in a manual.
Line 350: were
Line 363: in a negative mode
Conclusions: Please, do not use sentences from the other parts of the manuscript for the Conclusion section (e.g. the first sentences). It is necessary to re-write this section focusing on the conclusions and future perspectives of your studies. You can use the last paragraph to reformulate the section.
Lines 385-286: quorum sensing inhibitor – you can use directly the abbreviation introduced.
Lines 410-413: Please, read it again and remove the text that is not necessary (repetition).
References: I would like to recommend you to include all co-authors in the literature.
Line 458: Please, follow the instructions for literature formatting
Line 478: Food Res. Int.
Line 480: - an, remove e
Line 502: Ahmad.
Lines 505-506: Bacillus cereus in italics
Line 512: J. Food Biochem.
Supplementary file/s: I cannot open the archive.
Sincerely yours,
Reviewer
Reviewer 3 Report
The manuscript deals with the screening of quorum sensing inhibitor (QSI) from raw materials of Chinese herbs. The alcohol extract showed a strong QSI activity.
The QSI active compounds were determined by UPLC-MS/MS, and 21 compounds were identified. The principal components were organic acids, such as malic acid and succinic acid.
The work present several pitfalls. The weakest one is related to the chromatographic method. Several coelutins can be observed. It looks like that no proper method optimization was carried out. Further, for identification purposes an Orbitrap MS was employed. It is not unbelievable that some compounds are tagged as unknown e.g. peaks no 2 and 10. Also no quantification data are reported. The language is pretty poor and needs an effective polishing.
Reviewer 4 Report
The manuscript "The Campsis grandiflora flower polyphenols and its
organic acid components have inhibitory effects on
autoinducer type 1 Quorum Sensing" describes the activities of plant extract and to malic and succinic acids agaisnt bacteria.
Despite the study highlighted the polyphenols actions, malic and succinic acids were the major components of plant extracts.
The work is weak because did not use positive controls to compare the results obtaind with CFP extract.
The major weakness was the high concentration of MIC: "The minimum inhibitory concentration (MIC) of CFP was 1000 μg/ml". In this way, I think that the results are not relevant to be published unless as negative results to avoid anyone performing the sama experiments.
In order to consider the publication after a major review follows my suggestions to improve the study.
1) The results must be improved with an addition of experimets containing a positive control of knowing quorum sensing inhibitor (QSI) and antibiotics.
- what is the naringin concentration used by reference cited Liu et al[30]? consider used naringin or another control with MIC < 20 ug/mL.
2) Please consider expressing the results in a percentageof decrease instead of
ln 134-135: "When the CFP concentration increased from 100 to 400 μg/ml, the yields of violacein were
dramatically decreased,.."
3) ln 222: Please cite the reference to "And caffeic acid which has been reported to be QS inhibitor is the principal component." and consider used caffeic acid as positive control and in the association of CFP.
4) Table 2 needs format better. Verbascoside content represent 26% of the area chromatogrma and must be tested as QS inhibitor. Consider using standard commercial verbascoside.
If is not viable due to the value of comercial verbascoside consider a fractionation with butanol that concentrates polyphenols in neutral pH.
5) The authors claims that "Our experiment results show for the first time that malic..." "...acid and succinic acid have the inhibited effects on AI-1 quorum sensing of C. violaceum 026.
But the results (lines 232-233) show a high value of MIC: "The MIC of succinic acid and malic acid to C. violacium 026
are 1200 μg/ml and 800 μg/ml, respectively."
Please correct the spelling of C. violaceum instead C. violacium in the manuscript.
What is the reference compound has been considered in the literature as reference?
6) The secction "Conclusion" is not a good conclusion: in my opinion, the text contains the way of study and results. Consider revision in agreement with the authors guide of "Molecules" to rewrite.